# Domestic Accidents of Children in the Orodara District of Burkina Faso: Mothers’ Knowledge of First-Aid Practices

**DOI:** 10.3390/ijerph21050523

**Published:** 2024-04-24

**Authors:** Abou Coulibaly, Armel Emmanuel Sogo, Anata Bara, Barbara E. Wildhaber, Sophie Inglin

**Affiliations:** 1Biomedical and Public Health Department, Research Institute of Health Sciences, National Center for Scientific and Technological Research, Ouagadougou P.O. Box 7047, Burkina Faso; sogoarmel@gmail.com; 2Ministry of Health and Public Hygiene, Ouagadougou P.O. Box 7009, Burkina Faso; baraanata@yahoo.fr; 3Division of Pediatric Surgery, Department of Pediatrics, Gynecology and Obstetrics, University Hospitals of Geneva, University of Geneva, 1205 Geneva, Switzerland; barbara.wildhaber@hcuge.ch (B.E.W.); sophie.inglin@hotmail.com (S.I.)

**Keywords:** domestic accidents, mothers, children, knowledge, factors associated

## Abstract

First-aid practices after a domestic accident are not always known, especially in Africa. This study aimed to measure mothers’ knowledge of emergency procedures and attitudes in the event of domestic accidents in children. We conducted a cross-sectional study in the Orodara health district, Kénédougou province, Burkina Faso, among mothers of children aged 0–14 years. The dependent variable was the mothers’ knowledge of domestic accident first-aid practices, and the independent variables were the sociodemographic characteristics of the households and the mothers. Determinants were identified using linear regression with a threshold of 5%. A total of 798 mothers were surveyed. The mean knowledge score was 6.9 (standard deviation = 1.5) out of 19. Upon our multivariate analysis, the factors associated with the mothers’ knowledge about first-aid practices were the mothers’ age, the number of children under 14 years old living in the same household, the household size, the score for knowledge of non-recommended attitudes, the mothers’ level of education, and the place of residence. This study showed that awareness campaigns, especially in rural areas, seem important in improving mothers’ knowledge of first-aid practices in domestic accidents and, therefore, reducing the morbidity and mortality associated with domestic accidents.

## 1. Introduction

Domestic accidents occupy an important place in the world. According to the World Health Organization (WHO), in 2015, the injury-specific mortality rate for infants under five years of age was 73 per 100,000 population [1]. The injury death rate was reported at 29.6 per 100,000 infants aged 0 to 11 months and 32.7 per 100,000 for children aged 1 to 4 years, with road accidents and drowning being the main causes of death [2]. Among children aged over five years, the situation remains just as problematic: according to a study published in 2022, accidents are responsible for 32.9% of deaths among infants aged 5–19 years. Although a significant proportion of these accidents are due to road accidents (7.8% of all deaths), other types of accidents are no less significant (drowning 5.2% and self-inflicted injuries 4.2%) [3].

In light of these alarming findings, the WHO’s Sustainable Development Goal 3 (SDG3) focuses on healthy lives and well-being at all ages and sets targets for reducing premature mortality from maternal causes and injuries for the period 2016–2030 [4]. Similarly, the Global Strategy for Women’s, Children’s and Adolescent’s Health 2016-30 has survival as its main objective and adolescent mortality as one of its 16 key indicators [5]. In light of the various figures presented above, achieving SDG3, therefore, depends not only on implementing mechanisms to prevent injuries of all kinds but also on increasing the parents’ knowledge of recommended first-aid practices in children and adolescents suffering from domestic accidents.

The literature is characterized by a scarcity of studies on domestic accidents: they are underestimated, and very few scientific studies have assessed the level of parents’ knowledge regarding recommended actions and attitudes and those to be avoided in the event of domestic accidents in children. Worldwide, we found some studies that have examined mothers’ knowledge of first-aid practices regarding domestic accidents in Saudi Arabia and India [6,7]. The studies showed that mothers’ knowledge generally was low. Only 5.6% of the participants of a study conducted in Saudi Arabia [6] achieved a passing score, which was 12 correct answers out of 14 questions, and 58.2% of mothers had fair knowledge about first-aid in another study, also conducted in Saudi Arabia [7]. Similar results were also found in Lebanon, where a study revealed that 57.4% of participating mothers showed fair knowledge of child injury prevention [8]. The factors associated with mothers’ knowledge of first-aid practices regarding domestic accidents were the graduate level of education, having received formal first-aid training courses, and having previous experience of a child injury in the study conducted by Harere et al. in Saudi Arabia [6].

In Africa, studies conducted in Egypt showed that mothers’ knowledge of first-aid practices regarding domestic accidents is also low. The study conducted by Gouda et al. [9] revealed that 90.5% of mothers had unsatisfactory knowledge about home injuries in Egypt, and another study in Egypt showed that two-fifths of the mothers participating had fair knowledge regarding first-aid [10]. Regarding the factors associated with mothers’ knowledge, African studies revealed that (i) the mother’s age and the mother’s education level were statistically significant independent positive predictors of higher knowledge scores [9] in Egypt, (ii) age, education, and occupation were statistically significant independent positive predictors of knowledge scores in Egypt [10], and (iii) that age, area of residence, and the level of education of the participants played a variable role regarding first-aid awareness, attitude, and knowledge in Kinshasa, Democratic Republic of Congo [11] (but this knowledge covered first-aid in general and was not focused on domestic accident in children).

Lastly, these various studies showed that non-recommended actions were common in mothers’ practices regarding domestic accidents around the world as well as in Africa. For example, in India, potentially harmful substances, including lime, toothpaste, clay, and mud were used for burns [12], and in Saudi Arabia, 77.4% of the participants knew of and/or implemented traditional remedies as first-aid or to treat burns, with honey and toothpaste being the most common among these remedies (69.9% and 53.7%, respectively) [13]. In Ghana, burns also had the highest percentage (61%) of potentially harmful practices (e.g., applying kerosene) [14], and, in Zimbabwe, urine and crushed cockroaches were used after burn injuries [15].

Therefore, a few studies have been conducted in Africa, and these studies were carried out in Egypt [10,16,17] (where interventions were implemented and the effects on mothers’ knowledge were evaluated) and Ghana [18]. Apart from these few studies, no other research has been carried out on this topic.

In Burkina Faso, the Geneva University Hospitals, in partnership with the Ministry of Health and the Sanon Souro University Hospital in Bobo-Dioulasso, have initiated a project called the “Pediatric Surgery Development Plan”, with a community component focusing on preventing and managing pediatric victims of domestic accident. This project involves implementing various prevention strategies through radio broadcasts and awareness-raising sessions on domestic accidents, with the involvement of primary care providers (traditional practitioners and community workers). These awareness-raising actions regarding domestic accidents align with much-needed health promotion and could probably improve the population’s quality of life and well-being. Before implementing the interventions, we felt it essential to assess the current status of knowledge around the attitudes to be adopted in case of domestic accidents. Thus, this study aimed to measure mothers’ knowledge of emergency procedures and attitudes in the event of domestic accidents involving children aged 0–14 years and determine the associated factors.

## 2. Materials and Methods

### 2.1. Study Setting

This study was conducted in the Orodara health district in the province of Kénédougou, one of the seven health districts in the Hauts-Bassins region. It is located in the west of Burkina Faso. It has seven departments (Orodara, Koloko, Kangala, Kourinion, Banzon, Djigouéra, and Samogohiri), corresponding to seven communities, one of which is urban (Orodara). In 2022, the Orodara health district had one medical center with a surgical unit and 39 primary health centers, none of which had a particular security context.

### 2.2. Operational Definitions

Falls, burns, cuts, intoxications (medicinal or not), animal/insect bites, intentional injuries, and foreign bodies in the respiratory or digestive tracts were categorized as domestic accidents.

### 2.3. Study Population

Our study population included mothers with at least one child aged 0 to 14.

### 2.4. Type of Study

We performed a cross-sectional survey.

### 2.5. Primary Study Procedures

This primary study was a cross-sectional study. To calculate the size, the relevant indicator we used was the proportion of child victims of domestic accidents. The hypothetical prevalence was set to 25%, based on those recorded in Egypt and Ghana, which revealed a prevalence of domestic accidents of 20.6% [19] and 16% [20], respectively. We set the margin of error to 3% and the cluster effect to 2 (because we included several children per household). Therefore, we included approximately 805 households. We stratified the samples by area of residence, surveying 200 households in urban areas (the sectors of the town of Orodara) and 600 households in rural areas (the villages). In each household, only one woman was surveyed. The inclusion criteria were the following: (i) to have lived in the village for the previous six months, (ii) to have at least one child under the age of 14 years, and (iii) to give informed consent. We surveyed households in the 39 villages with a primary health center in the Orodara health district. Within each village, we surveyed a fixed number of households.

Data collection began in August 2022 and lasted three months. Over two days, we trained data collection staff on the questionnaire and survey techniques. During this training, all tools were translated into the local language (mainly Dioula) and pre-tested before the actual start of data collection. For each household sampled, the interviewers randomly selected a mother among those with a child under 14. The questionnaire was then administered to this mother, and her spontaneous responses were recorded. In other words, for each type of domestic accident, the woman was asked, for example, the following question: “What would you do if your child got burned?” Then, the interviewer recorded all the mother’s spontaneous answers. During data analysis, for each action recommended in the image box, used to raise awareness of domestic accidents, cited spontaneously by the mother, 1 point was awarded, and 0 points were awarded for the opposite case. So, we did not read the response options to the mothers and asked them to choose. All the interviews were carried out in the households. A data quality assurance plan was developed during the training phase. Implementation was under the supervision of the research team. Data were collected electronically using tablets. The confidentiality of the data was ensured by protecting the devices with passwords. The mothers in this research were reassured that their decision to participate would not affect how they were treated in the health centers. Moreover, they were given information about this study in a language of their choice (mainly in the Dioula language), in line with the ethical requirements of research involving human subjects. Those consented to participate in this study were asked to sign an informed consent form. Participants who could not read (or sign) put their fingerprints on the consent form (using an inkwell). Research team members’ telephone contacts were given to the parents should they require further information or assistance. The Burkina Health Research Ethics Committee approved this study (number 2022-08-182).

### 2.6. Variables

The dependent variable was the mothers’ knowledge of domestic accident first-aid practices (a discrete quantitative variable in the form of a score derived from 7 dimensions for 19 items). The independent variables mainly were the sociodemographic characteristics of the households and the mothers: mother’s age (coded in 4 categories: under 18; 18–24; 25–34; and 35 and over); mother’s level of education (none, primary, secondary, and higher); mother’s occupation (household, salaried worker or other income-generating activities, civil servant/pupil/student); household size (under 5 people, 5–9 people, 10+ people); number of children under 14 years of age in the same household; and place of residence (urban or rural).

### 2.7. Data Collection Sources, Tools, and Techniques

We used data extraction and interviews. The first version of the image box (Figure 1) designed by the Burkina Faso Ministry of Health and Public Hygiene and the Geneva University Hospitals on actions and attitudes toward domestic accidents for community use [21] inspired the questionnaire for assessing mothers’ knowledge. The questions included a list of recommended actions and attitudes for each type of domestic accident (7 questions, each with multiple choices). For each action recommended in the image box, one point was awarded for each correct answer given or 0 if not (spontaneous answers). The maximum score was, therefore, 19.

### 2.8. Data Management, Processing, and Analysis

For the analysis, data were exported to Stata version 18.0 for correction, data management, and analysis. The unit of analysis was the child’s mother. The search for determinants was carried out using linear regression with 5% as the significance threshold, given that the knowledge score variable was normally distributed. The variables were included in the model using a bottom-up stepwise procedure. The mean and standard deviation were calculated for quantitative variables such as the knowledge score, and the means were compared using Student’s *t*-test.

## 3. Results

### 3.1. Characteristics of Children’s Mothers

We surveyed a total of 798 mothers. The sample description shows that most mothers were housewives (73.4%), 58.4% of whom had never attended school. The mothers were mainly aged between 25 and 24 years (41.1%), had no education (58.4%), and were living in rural areas (74.2%). Detailed characteristics are shown in Table 1.

### 3.2. Knowledge of Recommended Actions and Attitudes in the Event of a Domestic Accident

Table 2 summarizes the mothers’ knowledge of recommended actions and attitudes in the event of a domestic accident. Overall, the mothers indicated that, in the case of a domestic accident, the recommended action was to send the child quickly to a primary health center for consultation. A total of 93.2% of mothers recommended this in the event of suffocation in the context of foreign body ingestion, 92.6% in the event of a burn, 90.9% in the event of a fracture, and 86.3% in the event of a bite.

For the most common domestic accidents, a large proportion of mothers cited a recommended practice. Yet, for these same injuries, 44.7% of mothers also cited at least one action/attitude that was not recommended and could be potentially dangerous, such as putting a tourniquet on a bitten child (5.4%), attempting suction on a bitten child (13.3%), giving liquids to intoxicated infants (17.9%), squashing a drowning child (14.9%), undressing a burned child (1.6%), or applying salt on burns (11.5%).

### 3.3. Mean Knowledge Score of Mothers and Factors Associated with Mothers’ Knowledge

The mean knowledge score was 6.9 (standard deviation = 1.5) out of 19. The factors associated with the mothers’ knowledge of first-aid practices in the event of a home accident are summarized in Table 3 (multivariate analysis).

## 4. Discussion

### 4.1. Mothers’ Knowledge of Domestic Accidents

We found that the mothers of infants had very little knowledge of the actions or attitudes recommended when faced with a domestic accident, with a mean score of only 6.9 out of 19. The only study carried out on the same subject shows that similar results were found in Egypt, in a study which showed that the mothers of infants gave an average of 11.0 correct answers to 29 questions about their knowledge, attitudes, and practices regarding the recommended actions in the event of a domestic accident involving children [17].

Indeed, the best-recommended attitude is still to consult a health center for the best management (cited by a large majority of mothers, with proportions ranging from 64.5% to 93.2%, depending on the type of domestic accident). However, this consultation must be carried out very quickly to avoid complications, and this is not always possible in African villages because the time to arrive to a health center and manage the condition is generally long [22,23,24,25]. Thus, mothers must know the first-aid practices necessary before arriving at the health center, as, for example, in the case of drowning or burns, where specific emergency procedures are required to avoid effects after the accident. Further, there are also specific actions that should not be performed so as not to aggravate the injury, as is the case, for example, in fractures, foreign body ingestion, or burns. However, we showed that these non-recommended actions are still commonly performed, with almost half of the mothers citing at least one non-recommended, potentially dangerous procedure. Similar results have been reported by previous studies, especially for burns, for which several authors mentioned potentially harmful actions. For example, in India, toothpaste, clay, and mud are frequently used for burns, as reported by Pathak et al. [12]; in Saudi Arabia, burns are often treated with honey and toothpaste [13,26]. Similar results were also found in Ghana, where, when providing first-aid for a fractured bone, a number of parents reported potentially harmful practices, such as seeking fracture care from a traditional bone setter or simply caring for the child at home. According to the authors, first-aid practices for burn care in Ghana also often included potentially harmful practices, ranging from applying petroleum jelly or raw eggs to the burn to applying kerosene [14]. Even in developing countries, mothers’ knowledge was low, such as in Greece, where only 10.5% of the participants would apply the optimal practice in the case of burns, consisting of rinsing the burn wound with cool running water for at least 10 min, applying only non-adhesive dressing on it and leaving the blisters intact [27].

### 4.2. Factors Associated with Mothers’ Knowledge

For the first time in the literature from the sub-Saharan region, we studied the factors associated with mothers’ knowledge of the recommended actions/attitudes when faced with a child suffering from a domestic accident. The results of our study showed that knowledge increased with the mothers’ age, the number of children under the age of 14 years living in the same household, the mothers’ education (secondary or higher), and, finally, their occupation as a civil servant, pupil, or student. The association between the mother’s age and her knowledge level could mean that the older the mother, the greater the likelihood that she has had to deal with a domestic accident involving one of her or a relative’s children. Also, the older she is, the more children she has under the age of 14 years, which also means a greater likelihood of having experienced a domestic accident involving a child. In fact, experience with domestic accidents could be the main reason for improved mothers’ knowledge. This has been reported by some authors, such as [7,28], although the study by Ho et al. [29] showed no association between previous domestic accidents and mothers’ level of knowledge. Another good explanation might be that mothers who have already brought another child involved in a domestic accident to a health center have learned the right attitudes and the ones to avoid, consequently increasing their knowledge scores. This supports our conclusion that education and awareness-raising actions are not only necessary but also, very probably, effective.

Interestingly, we also noted that the mothers’ knowledge of recommended behaviors decreased with the knowledge of non-recommended behaviors, rurality, and household size. In the case of rural areas, this result might be explained by the fact that mothers in these areas have less education than those in urban areas. As our results show, the mothers’ knowledge of recommended actions/attitudes increased with the level of secondary or higher education. However, these women were more likely to live in urban areas. Some authors in Egypt, such as Seifi et al., have also found an association between better knowledge of first-aid practices with age, level of education, and previous domestic accidents [16]. Eldosoky, also in Egypt, further showed that, besides the factors mentioned above, two more significant predictors of mothers’ good level of knowledge were (i) a better socio-economic status and (ii) participation in first-aid training, further reinforcing our conclusion that education and awareness-raising actions are efficient [17]. Other authors on other continents have also shown similar results, such as Johani in Saudi Arabia, who showed that the best-educated parents and those with experience with domestic accidents involving their children had better knowledge of first-aid procedures [28]. Wani et al., also in Saudi Arabia, found the same link between the level of education and the knowledge score [30] as Zedain et al. in Egypt [10].

Some limitations of this study may be situated in the data collection techniques (interviews): despite the interviewers being well-trained, the respondents still needed to answer the questions frankly. Given that certain actions and attitudes had often already been judged as being bad or good by health workers when the mothers had brought their infants involved in domestic accidents to health centers, the mothers might have been biased, and this may have influenced some of the mothers’ answers. Nevertheless, the collected data certainly give us a good idea of the mothers’ level of knowledge.

## 5. Conclusions

The results of our study show that the level of mothers’ knowledge of the first-aid practices to be adopted in the event of a child suffering from a domestic accident is very low. Mothers performing non-recommended actions highlights the widespread existence of poor beliefs and practices regarding the provision of first-aid to children having experienced domestic accidents. These findings suggest that educational programs are strongly needed to address the gaps in mothers’ knowledge of child injury prevention and first-aid measures and, finally, improve the situation in the long term. The first to be targeted should be mothers living in rural areas, given the strong link between the level of education and the knowledge of emergency practices. Further studies are needed on whether addressing fathers or other family members may help improve first-aid practices.

## Figures and Tables

**Figure 1 ijerph-21-00523-f001:**
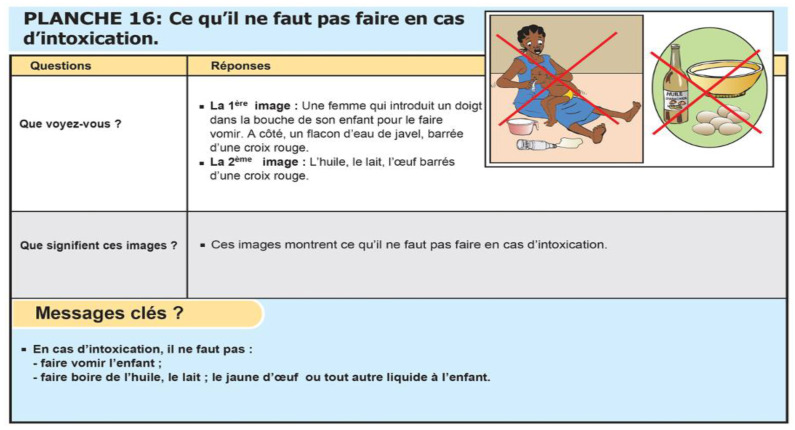
Extraction from the Burkinabe image box on domestic accidents [21] (printed with permission of the Ministry of Health and Public Hygiene of Burkina Faso). Translation: Question 1: “What do you see?” Reply: First image: a woman who introduces a finger in her child’s mouth to induce vomiting. On her side is a bottle of bleach. All crossed out with a red cross. Second image: a bottle of oil and bowl of milk. All crossed out with a red cross. Question 2: “What do these images mean?” These images show what one should not do in the event of intoxication. Key messages: In the event of intoxication, do not induce vomiting or make the child drink oil, milk, egg yolk, or any other liquid.

**Table 1 ijerph-21-00523-t001:** Characteristics of children’s mothers.

Mothers’ Characteristics	Total (n = 798)	Frequency (%)
Mother’s age (in years)		
Under 18	14	1.8
18–24	218	27.3
25–34	328	41.1
35 and over	238	29.8
Mother’s level of education		
None	466	58.4
Primary	163	20.4
Secondary and higher	169	21.2
Mother’s occupation		
Household	586	73.4
Salaried worker or other income-generating activities	185	23.2
Civil servant/pupil/student	27	3.4
Household size		
Under 5 people	382	47.9
5–9 people	362	45.4
10+ people	54	6.8
Number of under-14-year-old children in the same household		
1	252	31.6
2	277	34.7
3	167	20.9
4 or more	102	12.8
Place of residence		
Urban	206	25.8
Rural	592	74.2

**Table 2 ijerph-21-00523-t002:** Knowledge of recommended actions and attitudes in the event of a domestic accident.

Type of Domestic Accident	Recommended Actions and Attitudes (N = 798)	Total (n = 798)	Frequency (%)
Burn	Take the child immediately to the nearest health center	739	92.6
	Remove the child from the source of heat	24	3.0
	Cover the child with a loincloth or blanket in the event of a flame burn	15	1.9
	Cool the burned area with plenty of cool water	14	1.8
Fracture	Take the child immediately to the nearest health center	725	90.9
Bite	Go directly to the nearest health center with the child’s health record	689	86.3
	Wash the wound with soap	6	0.8
	Avoid using a tourniquet	1	0.1
Drowning	Evacuate quickly to the nearest health center	618	77.4
	Remove the victim from the water	427	53.5
	Place the child in the lateral safety position	2	0.3
	Press on the child’s chest to induce vomiting	28	3.5
Intoxication	Go directly to the nearest health center	663	83.1
	Take the toxic product to the health center for identification	58	7.3
Electroshock	Take the victim quickly to the nearest health center	515	64.5
	Cut off the power supply	141	17.7
Foreign body ingestion	Take the child quickly to a health center	744	93.2
	Place the infant flat on its stomach on a bent knee, face towards the ground	18	2.3
	Strike several times between the baby’s shoulder blades (at least 5 times), using the flat of your hand	83	10.4

**Table 3 ijerph-21-00523-t003:** Factors associated with mothers’ knowledge of recommended actions and attitudes in the event of a domestic accident.

Characteristics	Adjusted Coefficient	95% CI		*p*
Mother’s age (in years)	0.02	0.01	0.03	0.002
Number of children aged 0–14 years in the same household	0.15	0.05	0.25	0.005
Household size	−0.08	−0.14	−0.03	0.002
Score of knowledge of non-recommended actions/attitudes	−0.43	−0.52	−0.33	<0.001
Place of residence				
Urban	reference			
Rural	−0.66	−0.95	−0.38	<0.001
Mother’s level of education				
None	reference			
Primary	−0.04	−0.28	0.20	0.752
Secondary and higher	0.39	0.12	0.67	0.005
Mother’s marital status				
Single/separated	reference			
In union	0.23	−0.17	0.63	0.257
Mother’s occupation				
Household	reference			
Salaried worker or other income-generating activities	1.14	0.57	1.71	<0.001
Civil servant/pupil/student	0.05	−0.22	0.33	0.703

## Data Availability

The data presented in this study are available upon request from the corresponding author.

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
