# Peer review of "Domestic Accidents of Children in the Orodara District of Burkina Faso: Mothers’ Knowledge of First-Aid Practices"

_ijerph, 2024, doi:10.3390/ijerph21050523_

Round 1

Reviewer 1 Report

Comments and Suggestions for Authors

The outline of the theoretical background is concise, although somewhat limited: after providing the metadata, it focuses exclusively on the lack of African studies examined in the topic. If no relevant studies have been carried out in African countries on mothers' knowledge of first aid practices, it would be appropriate to refer to the results of studies conducted in other countries, and to highlight some thoughts about the four referred African studies. The literature sources used are adequate, but I would recommend further expansion.

The research is sufficiently delimited, and the presentation of the research objectives and sample selection is thorough. The text in Figure 1 should be translated into English. It would be advisable to include the applied measurement tool in its entirety in the study. The size of the sample is adequate, the presentation of the results is easy to follow.

In the discussion, the authors interpret the results in a social context. As the authors mentioned in the introduction that there were hardly any similar studies in Africa, it would be advisable to compare the results with studies carried out on other continents, where mothers with similar socioeconomic backgrounds were studied, and it would be possible to reflect on them when formulating conclusions and possible proposals. In line 194, references 16 and 17 should be incorporated more smoothly into the text.

The conclusions, which are somewhat rough, are well supported by the results and the discussion. However, further refinement and formulation of possible proposals would be useful. Suggestions concerning a broader social context could also be formulated (e.g. involvement of fathers and other family members, an increase in education would have a beneficial effect on this area in the long term, etc.).

Reviewer 2 Report

Comments and Suggestions for Authors

In this study, the authors assess maternal knowledge of first aid practices regarding domestic accidents in children in Burkina Faso. They explore this with a cross-sectional study designed using a survey that is loosely based on an existing survey from a Turkish research team. The introduction of the manuscript is well-written and clearly sets the stage (with only a few minor mistakes regarding the English language).

The materials and methods section of the manuscript needs additional clarification. The authors state that they included a study population of mothers with at least one child (aged between 0 and 14 years), but they failed to describe how these mothers were approached. Were they attending a healthcare clinic? When was this done (year, month)? Was informed consent provided by all study participants? Regarding the survey procedures, the authors write that they performed a power calculation which led them to include 800 mothers. Yet, they do not give any details on how this power calculation was performed.

My main critique is regarding the questionnaire used in this study. The authors refer to a Turkish study (reference 10) on which they based their own questionnaire. I had a detailed look at the other questionnaire, and it is very different from this one. The authors also do not describe if the local questionnaire was assessed by local experts for internal validation, nor do they state if a pilot experiment was performed to evaluate how well the questionnaire performed in this community.

Regarding data collection, I was intrigued to find that data were obtained using ‘extraction, interviews, and observation’. However, later in the text, I could not find which exact (numerical) data were collected. I assume that by ‘extraction, interview, and observation’, the authors mean that the study team approached participants verbally (due to the fact that many of these participants are illiterate). However, it is difficult to ensure that this is the correct interpretation based on how the manuscript is currently written.

In the results section, I had some difficulties with Table 2. The authors describe that they studied seven different types of accidents (burns, fractures, bites, drowning, intoxication, electrocution, and foreign body ingestion), which are assessed by a total of 19 questions. Based on the description, I am not sure if the different options provided for a specific type of accident are given to the participants, and if they have to choose which one they would do, or if the question is asked to them ("What would you do in case of a burn?") and they have to provide an answer. If the options are provided to the participants, I question why only one option is given for a fracture. It is indeed a good idea to bring a child with a fracture to a local health clinic; however, it is also important to immobilize, for instance, a fractured limb to prevent pain and further dislocation of fractured bone segments. Regarding foreign body ingestion, I do not understand why striking with a flat hand between the baby's shoulder blades is presented as a good option, unless it is referring to inhalation rather than ingestion, where a foreign body is stuck in the airways.

In the discussion, I do understand that there is very little comparative literature available on this topic, making it challenging to write a discussion comparing these data to other studies. Yet, sometimes the authors freely indulge in conjecture. Claiming that maternal age correlates with the knowledge level of first aid practices is one thing, but providing detailed explanations that this may be due to the fact that she may have experienced a domestic accident with a child before (and that this will increase her knowledge) is not evidence-based.

Despite my critical assessment, I applaud the energy and professionalism that the authors put into performing this study. However, I would prefer to see some changes made to the current manuscript before accepting it for publication.

Comments on the Quality of English Language

minor improvements necessary

Round 2

Reviewer 1 Report

Comments and Suggestions for Authors

The revision of the study was carried out well based on the suggestions, and the authors succeeded in further developing their manuscript. I recommend publishing the revised manuscript.

Reviewer 2 Report

Comments and Suggestions for Authors

the authors have addressed the reviewer comments adequately